# Impacts of Silver Nanoparticles on Plants: A Focus on the Phytotoxicity and Underlying Mechanism

**DOI:** 10.3390/ijms20051003

**Published:** 2019-02-26

**Authors:** An Yan, Zhong Chen

**Affiliations:** Natural Sciences and Sciences Education, National Institute of Education, Nanyang Technological University, Singapore 637616, Singapore; an.yan@nie.edu.sg

**Keywords:** plants, AgNPs, phytotoxicity, uptake, reactive oxygen species (ROS)

## Abstract

Nanotechnology was well developed during past decades and implemented in a broad range of industrial applications, which led to an inevitable release of nanomaterials into the environment and ecosystem. Silver nanoparticles (AgNPs) are one of the most commonly used nanomaterials in various fields, especially in the agricultural sector. Plants are the basic component of the ecosystem and the most important source of food for mankind; therefore, understanding the impacts of AgNPs on plant growth and development is crucial for the evaluation of potential environmental risks on food safety and human health imposed by AgNPs. The present review summarizes uptake, translocation, and accumulation of AgNPs in plants, and exemplifies the phytotoxicity of AgNPs on plants at morphological, physiological, cellular, and molecular levels. It also focuses on the current understanding of phytotoxicity mechanisms via which AgNPs exert their toxicity on plants. In addition, the tolerance mechanisms underlying survival strategy that plants adopt to cope with adverse effects of AgNPs are discussed.

## 1. Introduction

Due to their small size (between 1 and 100 nm) and unique chemical and physical characteristics, engineered nanomaterials (ENMs) were developed and expanded for application in many industrial sectors and daily life. Among various types of ENMs, silver nanoparticles (AgNPs) are the most commonly applied nanomaterial. It is reported that nearly 25% of all nanotechnology consumer products involve AgNPs [1]. Because of their well-known antibacterial and antifungal properties, they can be used in household products, food packaging, textiles, medical devices, antiseptics in healthcare delivery, and personal healthcare [2,3,4,5,6]. AgNPs can also be used in electronic devices and wastewater treatment because of their good electrical conductivity and photochemical properties [6,7].

In the agriculture sector, AgNPs were developed as plant-growth stimulators [8,9], fungicides to prevent fungal diseases [10], or agents to enhance fruit ripening [11,12]. The growing consumption of AgNPs inevitably increases the chance of release into the environment during AgNP synthesis and incorporation into products, as well as handling and recycling or disposal of these products [13,14,15]. AgNPs are expected to flow into environment as surface waters (e.g., lakes, streams, and rivers) [16], and the main pathway is through biosolids from wastewater treatment [17,18]. Indeed, AgNPs are detected widely in water and soil; they accumulate in the soil or water reservoirs in large quantities [19,20,21]. An analysis of the wastewater from a sewage treatment plant indicated existence of AgNPs with a size of 9.3 nm and a concentration of 1900 ng/L [22]. Moreover, the concentrations of AgNPs in surface water and sewage treatment are increasing significantly [21,23,24,25]. In agriculture, AgNP-contaminated water may permeate into fields through fertilization and irrigation [26]. The released AgNPs have the ability to permeate different media and eventually enter the plant rhizosphere [27,28]. Therefore, the AgNPs are inevitably taken up by crops and easily enter into the food chain [29], not only posing impacts on food production and food quality, but also posing a risk to human health [30,31,32,33].

Silver is the second most toxic metal to aquatic organisms after mercury [34]. Actually, AgNPs can leach silver ions (Ag^+^), which are persistent, bioaccumulative, and highly toxic to organisms [35]. Therefore, the release of AgNPs into ecosystems raises great concerns about their safety and environmental toxicity. As plants are a vital part of ecosystem and the primary trophic level in ecosystems, representing the base of the food chain [36,37], a good understanding of the impacts of AgNPs on plants is of paramount importance for assessing their toxicity [38]. Hence, the present review describes the uptake and translocation of AgNPs, and gives a detailed summarization of the impacts of AgNPs on plants. The phytotoxicity mechanisms via which AgNPs cause impacts on plants and the tolerance mechanisms through which plants alleviate the detrimental effects of AgNPs are discussed for a better understanding of interactions between plants and AgNPs.

## 2. Uptake and Translocation of AgNPs in Plants

In plants, AgNPs are transported via the intercellular spaces (short-distance transport) and via vascular tissue (long-distance transport) [29,39,40,41]. After exposure to plants, NPs penetrate cell walls and plasma membranes of epidermal layers in roots, followed by a series of events to enter plant vascular tissues (xylem), and move to the stele. Xylem is the most important vehicle in the distribution and translocation of NPs [42]. Through xylem, AgNPs can be taken up and translocated to leaves. In *Arabidopsis thaliana*, AgNPs can be taken up by the roots and transported to the shoots [29]. Geisler-Lee et al. found that AgNPs was taken up and progressively accumulated in the root tips, from border cells to root cap, epidermis, columella, and initials of the root meristem [39]. A further study indicated that AgNPs attached to the surface of primary roots in *Arabidopsis* and then entered root tips at an early stage after exposure. After 14 days, AgNPs gradually moved into roots and entered lateral root primordia and root hairs. After multiple lateral roots were developed, AgNPs were present in vascular tissue and throughout the whole plant from root to shoot [40].

The cell wall of the root cells is the main site through which AgNPs enter in plant cells [43]. In order to enter into the plant, AgNPs need to penetrate the cell wall and plasma membranes of epidermal layer of roots. The cell wall is a porous network of polysaccharide fiber matrices and, thus, acts as natural sieve [44,45]. The small-sized AgNPs can pass through the pores, whereas larger AgNPs are unable to enter into plant cells and are thereby sieved out [43].

Interestingly, AgNPs can induce the formation of new and large-sized pores, which permits the internalization of large AgNPs through the cell wall [44]. AgNPs can also be transported within the plant cell through the plasmodesmata process [29,46,47]. Plasmodesmata are pores of 50–60 nm in diameter and connect adjacent neighboring plant cells. In *Arabidopsis*, AgNPs are found to aggregate in plasmodesmata and in the cell wall [39], suggesting that there may be blockage of intercellular communication, which may be caused by the mechanical presence of AgNPs at these sites and may affect nutrient intercellular transport [40].

In addition to the root pathway, AgNPs can also be taken up through plant leaves. Geisler-Lee et al. found that if cotyledons of the *Arabidopsis* seedlings were immersed in AgNP-containing medium, AgNPs could be taken up and accumulated in stomatal guard cells [40]. Larue et al. found that AgNPs were effectively trapped on lettuce leaves by the cuticle after foliar exposure, and AgNPs could penetrate the leaf tissue through stomata [48]. In addition, Li et al. compared the uptake of AgNPs in soybean and rice following root versus foliar exposure, and found that foliar exposure resulted in 17–200 times more Ag bioaccumulation than root exposure [49].

Once the AgNPs enter into vascular tissues of crops, they can be taken up and transported to the leaves or other organs through long-distance transport [27,29,40]. Therefore, it is possible that the fruits, seeds, and other edible parts of plants may also be subjected to contamination by AgNPs through translocation.

## 3. Phytotoxicity of AgNPs

### 3.1. Phytotoxicity at the Morphological Level

After exposure to AgNPs, significant changes in the morphology of plants were observed. Growth potential, seed germination, biomass, and leaf surface area are the commonly used parameters for assessing the phytotoxicity of AgNPs in plants [27,42,43]. It was demonstrated that AgNP exposure could inhibit seed germination and root growth, and reduce biomass and leaf area. Jiang et al. found that AgNPs significantly decreased plant biomass, inhibited shoot growth, and resulted in root abscission in *Spirodela polyrrhiza* [50]. Kaveh et al. showed that exposure to higher concentrations (from 5 to 20 mg/L) of AgNPs resulted in reduction of the biomass in *Arabidopsis* [51]. Dimkpa et al. found that AgNPs reduced the length of shoots and roots of wheat in a dose-dependent manner in wheat [52]. Similarly, Nair and Chung showed that AgNPs significantly reduced root elongation, and shoot and root fresh weights in rice [53]. Stampoulis et al. demonstrated that AgNPs (>100 mg/L) inhibited seed germination and reduced biomass in zucchini *(Cucurbita pepo*) [54]. Similar results regarding the toxicity on seed germination, biomass accumulation, and root and shoot growth by AgNPs were reported in other studies involving various plant species, including *Arabidopsis* [55], *Brassica nigra* [56], *Lemna* [57], *Phaseolus radiatus* and *Sorghum bicolor* [58], *Lolium multiflorum* [59], rice [60], wheat [61], *Lupinus termis* L. [62], and so on. A summary of compiled descriptions of the effects of AgNPs in plants is shown in Table 1.

### 3.2. Phytotoxicity at Physiological Level

Phytotoxicity of AgNPs to plants at the physiological level is predicted by reduction of chlorophyll and nutrient uptake, decline of transpiration rate, and alteration of hormone. AgNPs can disrupt the synthesis of chlorophyll in leaves and, thus, affect the photosynthetic system of the plants [43]. Qian et al. showed that AgNPs could accumulate in *Arabidopsis* leaves, further disrupt the thylakoid membrane structure, and decrease chlorophyll content, leading to the inhibition of plant growth [55]. Nair and Chung reported that, after exposure to AgNPs for one week, total chlorophyll and carotenoids contents were decreased significantly in rice (*Oryza sativa* L.) seedlings [53]. Vishwakarma et al. found that AgNPs could accumulate in mustard (*Brassica* sp.) seedlings and caused severe inhibition in photosynthesis [71]. A recent study showed that AgNP exposure changed the thylakoid in *Physcomitrella patens*, and AgNPs decreased the chlorophyll b content and disturbed the balance of some essential elements in the leafy gametophytes [64]. In *Lupinus termis* L. seedlings, after exposure to AgNPs for ten days, the shoot and root elongation and fresh weights, total chlorophyll, and total protein contents were significantly reduced [62]. In *Cucurbita pepo*, the rate of transpiration was remarkably reduced after AgNP exposure [54,92,93].

In addition, AgNPs can affect the fluidity and permeability of the membrane and, consequently, influence water and nutrient uptake. Zuverza-Mena et al. demonstrated that AgNP exposure on radish (*Raphanus sativus*) sprout caused a decrease in water content in a dose-dependent manner; the nutrient content (Ca, Mg, B, Cu, Mn, and Zn) was also significantly reduced, suggesting that AgNPs may affect plant growth by changing water and nutrient content [76].

It was reported that AgNPs also affect plant hormones. Sun et al. found that the root gravitropism of *Arabidopsis* seedling was inhibited by exposure to AgNPs in a dose-dependent manner. Further analysis indicated that AgNPs reduced auxin accumulation, while gene expression analysis suggested that auxin receptor-related genes were downregulated upon AgNP exposure [70]. Vinković et al. conducted hormonal analysis using ultra-high-performance liquid chromatography electrospray, and found that AgNP accumulation in pepper tissue resulted in a significant increase in total cytokinin levels, suggesting the importance of cytokinin in the plant’s response to AgNPs stress [12]. Wang et al. found that Ag_2_S-NPs could reduce the growth of cucumber and wheat; expressions of six genes involved in ethylene signalling pathway were significantly upregulated in cucumber after exposure to Ag_2_S-NPs, suggesting that Ag_2_S-NPs could affect plant growth through an interface with the ethylene signaling pathway [72].

### 3.3. Cytotoxicity and Genotoxicity

AgNPs can also cause toxicity at the cellular and molecular level in plants. Many studies showed that the inhibition of plant growth after AgNP exposure is accompanied with alteration of cell structure and cell division. Yin et al. found that *Lolium multiflorum* seedlings failed to develop root hair, and the cortical cells were highly vacuolated and collapsed, while the epidermis and root cap were also damaged after exposure to 40 mg/L AgNPs [59]. Pokhrel and Dubey observed that AgNPs could reduce the size of the vacuole and lead to the reduction of cell turgidity and cell size in maize (*Zea mays* L.) and cabbage (*Brassica oleracea* var. *capitata* L.) [87,100]. Similarly, Mazumdar found that after AgNPs enter the cell of *Brassica campestris*; vacuoles and cell wall integrity were damaged, and other organelles might also be affected [63,101]. Likewise, Mirzajani et al. found that AgNPs with a concentration of to 60 µg/mL could penetrate the cell wall, and damage the cell morphology and its structure in rice [88]. In addition, Kumari et al. reported that AgNP exposure in *Allium cepa* significantly decreased the mitotic index and impaired cell division, resulting in chromatin bridge, stickiness, disturbed metaphase, multiple chromosomal breaks, and cell disintegration [98]. Similarly, Patlolla et al. demonstrated that AgNP treatment significantly increased the chromosomal aberrations and micronuclei, and decreased the mitotic index (MI) in root tip cells of broad bean (*Vicia faba* L.), suggesting that cell cycle and mitosis in root tip cells was disrupted by AgNPs [94]. A recent study confirmed that the root tip cells of wheat could readily internalize the AgNPs. After AgNP internalization, the root tip cells exhibited various types of chromosomal aberrations, such as incorrect orientation at metaphase, chromosomal breakage, spindle dysfunction, fragmentation, unequal separation, and distributed and lagging chromosomes, which seriously interfered with cell function [63]. The uptake, translocation, and major phytotoxicity of AgNPs in plants are illustrated in Figure 1.

## 4. Toxicity Mechanisms

### 4.1. AgNP-Induced Oxidative Stress

The main mechanism underlying the phytotoxicity of AgNPs is the production of excess reactive oxygen species (ROS) induced by AgNPs, resulting in oxidative stress in plant cells [100,103]. A number of studies demonstrated that ROS production is significantly elevated in plants after exposure to AgNPs. There are four types of ROS produced in plant cells, including singlet oxygen (^1^O_2_), superoxide (O_2_^•−^), hydrogen peroxide (H_2_O_2_), and hydroxyl radical (HO^•^) [36,104]. Under normal environmental conditions, ROS are generated as byproducts of normal metabolic pathways in organelles such as chloroplasts, mitochondrion, and peroxisomes [36,105]. Under stressed conditions, however, excessive amounts of ROS are generated and cause severe oxidative damage to plant biomolecules through electron transfer [106]. The production of excess ROS induced by AgNP exposure can subsequently lead to oxidative stress, cause peroxidation of polyunsaturated fatty acids (known as lipid peroxidation), and damage the cell membrane permeability and alter cell structure, directly damaging protein and DNA, resulting in potential cell death and growth inhibition in plants (Figure 1) [36,100,107,108,109]. For example, Panda et al. reported that AgNP-P (phyto-synthesized from silver nitrate AgNO_3_) or AgNP-S (commercial AgNPs from Sigma–Aldrich) application in *Allium cepa* significantly increased the generation of superoxide (O_2_^•−^) and H_2_O_2_; they also induced cell death to different extents in a dose-dependent fashion, following an order of AgNP-S > AgNP-P at doses ≥20 mg/L. Moreover, AgNP-P significantly decreased the mitotic index. Comet assay suggested that DNA damage was significantly enhanced after AgNP-P and AgNP-S treatments in a dose-dependent manner, whereby AgNP-S (threshold dose ≥ 10 mg/L) is more genotoxic than AgNP-P (threshold dose ≥ 20 mg/L) [97]. Qian et al. found that AgNPs could accumulate in *Arabidopsis* leaves and change the transcription of antioxidant and aquaporin genes, suggesting that AgNPs can change the balance between oxidant and antioxidant systems [55]. Similarly, Speranza et al. checked the in vitro toxicity of AgNPs to kiwifruit pollen, and found that changes in ROS generation paralleled the entire germination dynamics of kiwifruit pollen. The AgNP treatment delayed H_2_O_2_ production, whereas AgNPs dramatically induced ROS overproduction at the late stage during pollen germination, leading to decreases in pollen viability and performance [110]. Moreover, Torre-Roche et al. found that AgNP exposure with concentration at 500 and 2000 mg/L caused significant increases (54–75%) in malondialdehyde (MDA) formation in soybean (*Glycine max*) [111]. MDA is a major peroxidation product under stress conditions and is indicative of the extent of lipid peroxidation [112]. Similarly, Nair and Chung reported that lipid peroxidation increased significantly after exposure to 0.2, 0.5, and 1 mg/L AgNPs in *Arabidopsis* [84]. In rice, Nair and Chung found that exposure to 0.5 and 1 mg/L AgNPs resulted in a significant increase in H_2_O_2_ formation and lipid peroxidation in shoots and roots; further analysis suggested that ROS production was promoted by AgNPs in a dose-dependent manner [53]. Thiruvengadam et al. reported the impact of AgNP exposure in turnip seedlings, and found that a higher concentration of AgNPs caused excessive generation of superoxide radicals and increased lipid peroxidation; H_2_O_2_ formation was also significantly increased after exposure to 5 and 10 mg/L AgNPs. Dichlorofluorescein (DCF) fluorescence indicated a sharp increase in ROS production in turnip seedling roots, suggesting the existence of oxidative stress in the roots after AgNP exposure. Further analysis by comet assay and terminal deoxynucleotidyl transferase-mediated dUTP nick end labeling (TUNEL) assay confirmed that DNA damage was significant, suggesting that AgNPs can induce cell death through apoptosis [113].

### 4.2. Silver-Specific Toxicity

It was shown that AgNPs can leach ionic silver (Ag^+^) into the surroundings through the oxidation of zero-valent Ag [114]. During AgNP uptake and translocation, Ag^+^ is released from AgNPs, resulting in oxidative stress through the generation of ROS and disturbing cell function, causing phytotoxicity by binding to cell components and modifying their activities [115,116,117]. For example, Speranza et al. analyzed the ion release kinetics of AgNPs in the pollen culture medium, and found that AgNPs rapidly dissolved into ions and reached a maximum of 11.8 wt.% ion release. The released Ag^+^ caused a fivefold increase in H_2_O_2_ production over controls; moreover, the released Ag^+^ damaged pollen membranes and inhibited germination to a greater extent than the AgNPs themselves, suggesting that Ag^+^ may excert its impacts mostly through chemical or physicochemical interactions with nucleic acids to induce DNA damage [110]. A gene expression study by microarray in *Arabidopsis* compared gene expression profiles between AgNP and silver ion (Ag^+^) treatments and found a significant overlap between genes differentially expressed in the two treatments, suggesting a similarity between plants’ responses to AgNPs and Ag^+^ [51]. Actually, when AgNPs oxidize in water, they can make bonds with anions and transform into the characteristics of heavy metals, which is more hazardous [43,118]. It was demonstrated that the conversion of AgNPs to a complex of anion or heavy metal could cause toxic effects on various living organisms [25,119,120,121].

Ag^+^ can also affect photosynthesis through competitive substitution of Cu^+^ in plastocyanin (Pc). Pc is a soluble copper-binding protein found in the thylakoid lumen of the chloroplast. It functions as an electron carrier to transfer electrons from cytochrome *b_6_/f* to photosystem 1 (PS1) in the photosynthetic electron-transfer (ET) chain [122,123]. Pc contains a type 1 copper site, where the copper ion is surrounded by two histidine ligands (His87, His37) and a cysteine ligand (Cys84) [124]. Ag^+^ can competitively replace Cu^+^ and bind to Pc, which results in disturbance or inactivation of the photosynthetic electron transport. Sujak found that Ag-substituted Pc occupied the active Pc electron transfer site of the cytochrome *f*, and caused a decrease in the turnover of the cytochrome complex [125]. Similarly, Jansson and Hansson demonstrated that Ag(I)-substituted Pc competitively inhibited electron transfer between normal Cu-containing Pc and PS1 [126]. Since both AgNPs and dissolved silver can be toxic to plants, the phytotoxicity of AgNPs becomes complicated, as the plant is subjected to both silver-specific and nanoparticle-specific biological effects [51]. Therefore, it is difficult to say whether the phytotoxicity is caused by ionic sliver or by intrinsic properties of AgNPs in certain AgNP application scenarios.

### 4.3. AgNP-Specific Toxicity

Although the phytotoxicity of AgNPs was associated with the impact of dissolved Ag^+^ on plants, the phytotoxicity effect could not be explained solely by the activity of the released Ag^+^ ions. In some case, AgNPs can be even more toxic than free Ag^+^ ions even at the same concentrations of Ag^+^ [127]. In another study, AgNP exposure to *Cucurbita pepo* caused more reduction in biomass and transpiration when compared with bulk Ag [93]. These studies suggest that free Ag^+^ ions contribute only partially to the phytotoxicity of AgNPs, while the intrinsic properties of AgNPs are critical for the phytotoxicity of AgNPs.

Indeed, the physical interactions between AgNPs and plant cell-transport pathways can influence the phytotoxicity of AgNPs [29]. Uptake of AgNPs into plant tissue may cause inhibition of apoplastic trafficking by clogging of pores and barriers in the cell wall or the nano-sized plasmodesmata, thereby effectively inhibit the apoplastic flow of water and nutrients [43,128].

A number of studies suggested that the effect and phytotoxicity of AgNPs are closely associated with the nature of the interactions between plants and AgNPs, which are determined by the intrinsic properties of AgNPs [100,129]. These physical and chemical properties of AgNPs, including size, shape, exposure concentration, surface coating, Ag form, and aggregation state, greatly influence the effect of AgNPs on different aspects of plant morphology, physiology, and biochemistry [130].

Among these properties, the size of AgNPs is critical for the phytotoxicity of AgNPs [129]. The smaller-sized AgNPs have a larger surface area to mass ratio, which allows better interference with cell membrane function by directly reacting with the membrane. Meanwhile, a higher proportion of the atoms of the particle on the surface can affect the interfacial reactivity and the ability to pass through physiological barriers [129,131]. It was shown that smaller AgNPs could accumulate to higher levels in plants and be more toxic than their bulk particles. Geisler-Lee et al. checked the impact of AgNPs with different sizes (20, 40, and 80 nm) on *Arabidopsis*, and found that smaller AgNPs accumulated more in seedlings than larger AgNPs (20 nm > 40 nm > 80 nm) at low concentrations. Moreover, smaller-sized AgNPs had a greater impact on root browning [39]. In another study, Wang et al. reported that smaller AgNPs (5 and 10 nm) accumulated to higher levles in poplar tissues than the larger 25-nm AgNPs when applied within the particle subinhibitory concentration range; both *Arabidopsis* and poplar showed susceptibility to the toxic effects of AgNPs, and this susceptibility increased with decreasing AgNP size [90]. Various phytotoxicity studies using different sizes of AgNPs suggested that phytotoxicity is negatively correlated with the size of AgNPs, as AgNPs with smaller size are generally more toxic to the plants than larger AgNPs [33,83,91,130]. For example, Yin et al. showed that AgNP toxicity was influenced by AgNP surface area; smaller AgNPs (6 nm) more strongly affected plant growth than larger (25 nm) AgNPs when applied with similar concentrations in *Lolium multiflorum* [59]. Similarly, another study showed that 6-nm gum arabic coated silver nanoparticles (AgNP-GA) have stronger effects on germination and growth of wetland plants than 21-nm polyvinylpyrrolidone-coated silver nanoparticles (AgNP-PVP) [91]. Abdel-Azeem and Elsayed examined the effect of different sizes of AgNPs (20, 50, and 65 nm) on *Vicia faba* and found that the effect of AgNPs on the mitotic index and chromosomal aberrations was AgNP size-dependent, as smaller-sized AgNPs caused a lower mitotic index and root growth values, confirming that smaller AgNPs are more toxic to *Vicia faba* [132].

Although these studies demonstrated that smaller AgNPs cause more phytotoxicity than larger AgNPs, this correlation between AgNP size and phytotoxicity is not always true for every AgNP exposure scenario. For example, Thuesombat et al. examined the effects of different sized AgNPs (20, 30–60, 70–120, and 150 nm) on seed germination and seedling growth in jasmine rice (*Oryza sativa* L. cv. *KDML 105*), and found that smaller AgNPs accumulated to higher levels than larger AgNPs, which is consistent with previous studies. However, both seed germination and seedling growth were decreased with increasing size; the 20-nm AgNPs treatment resulted in the less negative effects on seedling growth when compared to treatment with the larger AgNPs (150 nm), which is contrary to previous reports. Further analysis found that 20-nm AgNPs were trapped in the roots rather than transported to the leaves, thereby causing less phytotoxicity on seedling growth than 150-nm AgNPs [86].

Numerous studies on the phytotoxicity of AgNPs revealed that the phytotoxicity of AgNPs is positively correlated with the concentration of AgNPs during exposure. AgNPs can only cause negative effects on plants when applied with a concentration above a certain threshold. Mirzajani et al. showed that AgNPs were unable to change cell morphology or structure of rice root when present in low concentrations (30 µg/mL), whereas, with an increased concentration of 60 µg/mL, AgNPs not only penetrated the cell wall, but also destroyed the cell morphology and the structural features. Moreover, 30 μg/mL AgNPs even accelerated root growth, while AgNPs at 60 μg/mL restricted root growth [88]. Oukarroum et al. reported that AgNP treatment induced intracellular ROS production in the aquatic plant *Lemna gibba*; the induced oxidative stress was positively correlated with the increasing concentration of AgNPs [133]. Similarly, Thuesombat et al. showed that seed germination and subsequent seedling growth were decreased with increased concentrations of AgNPs in jasmine rice [86]. Cvjetko et al. found that AgNPs induced oxidative stress and exhibited phytotoxicity only when applied in higher concentrations in *Allium cepa* roots [67].

Engineered AgNPs are typically stabilized against aggregation through surface coating, using organic or inorganic compounds to coat the surface of AgNPs to obtain electrostatic, steric, or electrostatic repulsive forces between particles [134]. Surface coating may change AgNP properties such as optical properties, dispersion, and shape [65,135], thereby influencing the toxicity of AgNPs to plants. Cvjetko et al. compared the toxicity of three types of AgNPs with different surface coatings (citrate, polyvinylpyrrolidone (PVP), and cetyltrimethylammonium bromide (CTAB)) on *Allium cepa* roots, and found that plants treated with AgNP-CTAB had significantly higher Ag content than plants treated with AgNP-citrate and AgNP-PVP, leading to strong inhibition of root growth and oxidative damage. Among the treatments of three types of AgNPs, AgNP-CTAB caused the highest toxicity, whereas AgNP-citrate showed the weakest effects, as AgNP-citrate was much bigger in size and aggregated to larger particles. These observations suggest that the toxicity of AgNPs is correlated with the size and surface coating [67]. Similarly, Pereira et al. found that AgNP-PVP was more deleterious on the growth rate and fronds per colony than AgNP-citrate in *Lemna minor*, whereby AgNP-PVP reduced the growth rate 1.5-fold more than AgNP-citrate [65]. In another study, Liang et al. observed the responses of *Physcomitrella patens* to AgNPs with different surface coatings at the gametophyte stages, and found that AgNPs without surface coating caused the worst damage to the chlorophyll of protonemata, whereas AgNP-PVP and AgNP-citrate just displayed negligible influence, suggesting that surface coating alleviated the damage of AgNPs to the chlorophyll of protonemata. However, at the leafy gametophyte stage, exposure to AgNP-citrate led to the highest weight loss of leafy gametophytes, followed by AgNP-PVP and AgNPs without surface coating [64]. These observations suggest that the effects of AgNPs with different surface coatings on plants are complicated and are associated with the stability of AgNPs, as well as different plant systems.

In addition, the morphology of AgNPs also influences the effect of AgNPs on plants. Syu et al. studied the impacts of AgNPs with three different shapes (spherical, decahedral, and triangular) on *Arabidopsis*, and found that decahedral AgNPs induced the highest degree of root growth promotion but the lowest levels of Cu/Zn superoxide dismutase (CSD2) accumulation. Triangular AgNPs also enhanced root growth, whereas spherical AgNPs exhibited no root growth promotion, but induced the highest levels of anthocyanin and CSD2 accumulation, suggesting that different morphologies of AgNPs exhibited different levels of effects on *Arabidopsis* [85]. A schematic diagram of AgNPs-specific toxicity is shown in Figure 2.

Based on various studies on the phytotoxicity of AgNPs, it is evident that the interaction between plants and AgNPs is highly complicated and is not only dependent on the intrinsic properties of AgNPs, but is also influenced by plant species, developmental stages, different tissues, and sample preparation methodologies.

## 5. Tolerance Mechanisms

Phytotoxicity of AgNPs is highly associated with oxidative stress, which is caused by the production of excess amounts of ROS after AgNP exposure. To avoid the detrimental effects of ROS, a set of antioxidant defense mechanisms are activated in plant cells. The defense mechanism involves the activities of enzymatic antioxidants such as superoxide dismutase (SOD), catalase (CAT), ascorbate peroxidase (APX), guaiacol peroxidase (GPX), dehydroascorbate reductase (DHAR), and glutathione reductase (GR) [100,136]. As different types of ROS have different modes of action and exhibit different effects on cellular organelles of plant cells, they can be balanced or removed by specific antioxidant enzymes [36,137]. For example, there are three types of SOD in plant cells, including Fe-SOD, Mn-SOD, and Cu-Zn-SOD, and they can rapidly convert highly toxic ROS (O_2_^•−^) to less toxic species (H_2_O_2_). CAT can convert H_2_O_2_ to H_2_O and O_2_. APX is able to convert H_2_O_2_ to H_2_O via ascorbate oxidation into monodehydroascorbate (MDA) and dehydroascorbate (DHA), both of which can be recycled to produce more ascorbate via the catalysis of MDA reductase (MADR) and DHAR [36]. Upon exposure to AgNPs, activities of these enzymatic antioxidants are elevated in plant cells to protect the cells from oxidative stress. For example, Zou et al. observed obvious oxidative damage to *Wolffia globosa* when the plants were exposed to 10 mg/L AgNPs. Meanwhile, the SOD activity was increased by 2.52 times, suggesting that the ROS-scavenging mechanism was activated [79]. Similarly, elevated SOD activity was also observed after AgNP exposure in tomatoes (*Lycopersicon esculentum*) [89]. Enhancement of peroxidase and catalase activity was also observed in *Bacopa monnieri* (Linn.) after AgNP treatment [74,83,138]. Jiang et al. found that the catalase activity in cells of *Spirodela polyrhiza* was significantly increased. Moreover, the SOD and peroxidase activity, and the antioxidant glutathione content were increased in a dose-dependent manner after exposure to 6-nm AgNPs [83]. In addition, Bagherzadeh Homaee and Ehsanpour examined the effects of AgNPs on potato (*Solanum tuberosum* L.) and observed that the activities of SOD, CAT, APX, and GR were all significantly increased in AgNP-treated plantlets [74].

Non-enzymatic antioxidants, such as anthocyanin, ascorbate, glutathione, and thiols, also contribute to the antioxidant defense mechanisms [100,136]. Anthocyanin is a kind of pigment that is implicated in tolerance to various biotic or abiotic stresses, such as herbivores and pathogens, drought, cold, ultraviolet (UV) radiation, and heavy metals [139]. Anthocyanin commonly serves as a non-enzymatic antioxidant to scavenge free radicals and chelate metals under stress conditions [36,104,106,139]. It was reported that anthocyanin accumulation was significantly induced in the spherical AgNP-treated *Arabidopsis* seedlings and was dose-dependent [85]. Similarly, anthocyanin accumulation was also significantly increased after exposure to higher concentrations of AgNPs in turnip [113]. In addition, other antioxidants such as ascorbic acid, carotenoids, and proline are also implicated in antioxidant defense responses of plants to AgNPs. Carotenoids are able to induce antioxidant activity and potentially reduce the toxic effects of ROS [140,141]. After AgNP exposure, a large increase in shoot carotenoid content was observed in rice, suggesting that plants employ carotenoid to reduce the effects of ROS caused by AgNPs [88]. An increase in ascorbic acid content was observed in *Asparagus officinalis* [99].

At the molecular level, the expression changes of genes that are associated with the response to AgNPs may underlie the antioxidant defense mechanisms of plants in response to AgNPs. Dimkpa et al. checked the transcription of a gene encoding metallothionein (MT), which is a cysteine-rich protein involved in detoxification by metal ion sequestration, and found that the expression of this gene was highly induced after AgNP treatment in wheat (*Triticum aestivum* L.) [52]. A gene expression study by microarray suggested that AgNP exposure to *Arabidopsis* led to the upregulation of genes that are associated with response to metal and oxidative stress, including genes encoding SOD, cytochrome P-450-dependent oxidase, and peroxidase, whereas AgNP exposure caused the downregulation of genes that are involved in response to pathogens and hormonal stimuli [51]. In *Arabidopsis*, the expressions of sulfur assimilation, glutathione biosynthesis, glutathione *S*-transferase, and glutathione reductase genes were significantly upregulated after exposure to AgNPs [84]. Sulfur metabolism in plants plays an important role in stress tolerance, especially in metal detoxification and in the maintenance of cellular redox homeostasis [117,142]. In addition, exposure of AgNPs to rice seedlings led to the differential transcription of genes associated with oxidative stress tolerance in shoots and roots, such as *FSD1*, *MSD1*, *CSD1*, *CSD2*, *CATa*, *CATb*, *CATc*, *APXa*, and *APXb* [53].

## 6. Potential Risk in Human Health Posed by AgNPs via Food Chain

Plants are producers in the ecosystem and represent the primary trophic level in the food chain. Regarding the food safety issue, most of the harvested edible tissues or organs of vegetables or cereals are consumed by livestock and humans. Since AgNPs can be taken up and accumulated in plants, they can further pose a risk to human health through invading the food chain and ultimately transferring to the human body. Actually, it was demonstrated that AgNPs could cycle in the ecosystem through various trophic levels in an aquatic or terrestrial food chain [9,100,143,144].

In aquatic ecosystems, planktonic algae as primary producers are located at the base of the aquatic food chain; therefore, algae were selected as the basic trophic level to investigate trophic transfer of AgNPs in a few studies. McTeer et al. investigated the bioavailability, toxicity, and trophic transfer of AgNPs between the alga *Chlamydomonas reinhardtii* and the grazing crustacean *Daphnia magna*, which belong to two different trophic levels. Nano Ag derived from AgNPs was accumulated into microalgae. After feeding on Ag-containing algae, *Daphnia magna* accumulated nano-derived Ag, confirming the trophic transfer of AgNPs between algae and *Daphnia magna* [145]. Similarly, Kalman et al. studied the bioaccumulation and trophic transfer of AgNPs in a simplified freshwater food chain comprising the green alga *Chlorella vulgaris* and *Daphnia magna*. After AgNPs were accumulated in algae, the Ag-contaminated algae were fed to *Daphnia magna*. Ag uptake in *Daphnia magna* was observed a few days later. Further analysis indicated that diet is the dominant pathway route of Ag uptake in *Daphnia magna* [144]. In addition, a recent study used paddy microcosm systems to estimate the trophic transfer of AgNP-citrate and AgNP-PVP among various trophic level organisms (aquatic plants, biofilms, river snails, and Chinese muddy loaches). After exposure, AgNPs rapidly coagulated and precipitated on the sediment. Stable isotope analysis indicated a close correlation between the Ag content in the prey and that in their corresponding predators, demonstrating the impact of AgNPs on ecological receptors and food chains [146].

In terrestrial food chains, studies on the potential trophic transfer of AgNPs remain scarce. However, the terrestrial trophic transfer of other metallic nanoparticles was investigated, such as AuNPs [147], CeO_2_-NPs [148], and La_2_O_3_-NPs [149]. In a simulated terrestrial food chain, tobacco hornworm (*Manduca sexta*) caterpillars were fed tomato leaf that were surface-contaminated with AuNPs. Later, the transfer of AuNPs from tomato to tobacco hornworm was observed [147]. Hence, these studies imply a possibility that AgNPs may also be transferred in the terrestrial food chains.

Both in vivo and in vitro studies demonstrated the toxicity of AgNPs on mammalian cells. For example, AgNP exposure reduced lung function and produced inflammatory lesions in the lungs of rat [150], and resulted in the accumulation of AgNPs in the olfactory bulbs and in the brain of rats [151]. Since AgNPs can be accumulated and transferred in the food chain, they may become dangerous to humans. Indeed, AgNPs exposure to human cells can stimulate inflammatory and immunological responses, cause oxidative stress, and lead to cellular damage [152,153,154]. Therefore, there is an urgent need to increase our understanding of the bioaccumulation and trophic transfer of AgNPs in the food chain, which is critical for assessing and mitigating their potential harm to human health.

## 7. Conclusions and Perspectives

Due to the immense application of AgNPs in various fields in modern society, their dispersal and permeation into the ecosystem became inevitable. Hence, a great concern is arising related to the potential risk of destruction in the ecosystem, decline in food quality and yield, and even undermining of human health imposed by AgNPs. To this concern, understanding how AgNPs transfer through the ecosystem and exert impacts on plants is of crucial importance. During the past decade, the research communities undertook the responsibility to increase our knowledge of the impacts of AgNPs on plants, by carrying out numerous studies regarding the interactions between plants and AgNPs. Most of these studies revealed the detrimental effects of AgNPs on plants in various aspects, including at morphological, physiological, cellular, and molecular levels. However, a few studies reported the positive effects of AgNPs on plant growth and development. These contradictory results indicate the complexity of the responses of plants to AgNPs, which are not only determined by the properties of AgNPs (size, concentration, shape, surface coating, Ag chemical form, etc.), but are also dependent on the plant system used (species, tissue, organ, developmental stage, etc.) and experimental methodology (medium, exposure method, exposure time, etc.)

In response to AgNPs, it is rational that multiple detoxification strategies may be activated; different plant species may employ different detoxification mechanisms to eliminate the toxic effects of AgNPs. Therefore, it is difficult to make a general conclusion on how different detoxification pathways in response to diverse AgNPs conditions are activated in different plant species. To address this issue, it is necessary to use representative species, such as the commonly used model plant *Arabidopsis*, to evaluate the phytotoxicity of AgNPs and tolerance mechanisms. Meanwhile, the establishment of a standardized methodology is required to conduct normalized AgNP exposure, thereby allowing comparisons between different species.

Although joint efforts by research communities generated essential knowledge of the impacts of AgNPs on plants, most of these experimental outcomes were based on laboratory experiments under controlled conditions that are likely far from field conditions, such as the exposure method (hydroponic vs. soil), exposure dosage, and time (acute vs. chronic). Therefore, it is hard to predict whether the phytotoxicity of AgNPs and tolerance mechanisms under laboratory conditions are the same as under field conditions. To this end, the establishment of well-designed, plant life-cycle experimental systems under environmentally realistic conditions is required to accurately evaluate the impacts of AgNPs on plants and to generate environmentally relevant implications.

In addition, most studies performed during the last decade focused on the impacts of AgNPs on plants at the morphological and physiological levels; however, the profound impacts of AgNPs at the molecular level did not draw enough attention. Benefits from the development of systems biology and multiple omics methodologies, such as transcriptomics, proteomics, and metabonomics, can be employed in future studies to comprehensively assess the phytotoxicity mechanism of AgNPs and tolerance mechanisms in plants.

## Figures and Tables

**Figure 1 ijms-20-01003-f001:**
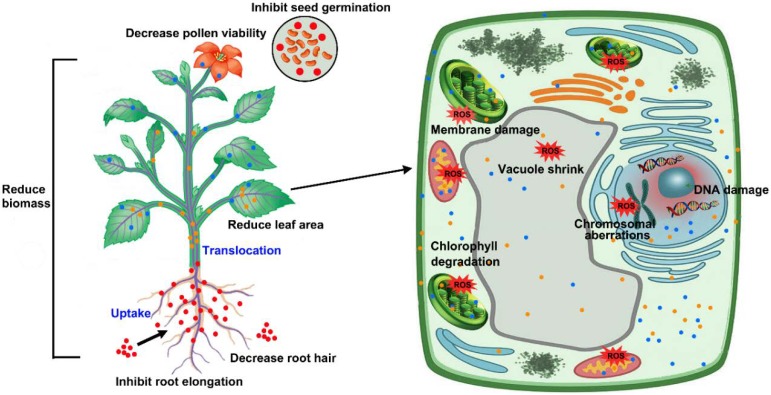
Schematic diagram representing uptake, translocation, and major phytotoxicity of silver nanoparticles (AgNPs) in plant (modified from Reference [102]). Generally, AgNPs are taken up by underground tissues (primary roots and lateral roots), then translocated to aboveground parts (stem, leaf, flower, etc.), where they can reduce biomass, decrease leaf area, affect pollen viability, and inhibit seed germination. At the cellular level, AgNPs enter into various organelles, leading to the production of excess reactive oxygen species (ROS), thereby causing cytotoxicity and genotoxicity, such as membrane damage, chlorophyll degradation, vacuole shrinkage, DNA damage, and chromosomal aberrations.

**Figure 2 ijms-20-01003-f002:**
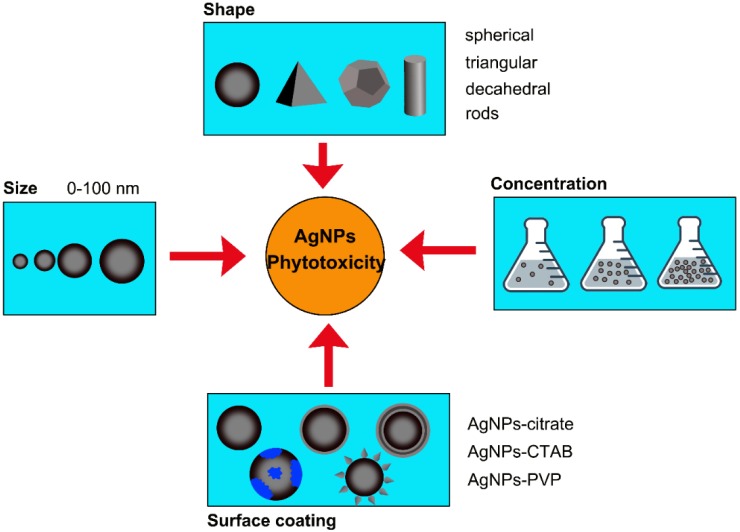
Schematic diagram showing AgNP-specific toxicity. The phytotoxicity of AgNPs is determined by AgNP properties, including size, shape, concentration, and surface coating of AgNPs.

**Table 1 ijms-20-01003-t001:** Summary of studies on phytotoxicity of silver nanoparticles (AgNPs) in plants.

Size (Diameter in nm)	Concentration	Species	Impacts	References
25–70; 7.5–25.0	10, 20, 40, 50 ppm	Wheat (*Triticumaestivum* L.)	Caused various types of chromosomal aberrations	[63]
5–10	0, 0.1, 0.3, 0.5 mg/L	*Lupinus termis* L.	Reduction in shoot and root elongation, shoot and root fresh weights, total chlorophyll, and total protein contents; Decreased sugar contents and caused significant foliar proline accumulation; Caused metabolic disorders	[62]
37.4 ± 13.4 (AgNP-B ^a^); 29.0 ± 6.0 (AgNP-PVP ^b^); 21.5 ± 4.2 (AgNP-Citrate)	5, 10 μg/mL	Bryophyte (*Physcomitrella patens*)	Inhibited the growth of the protonema; Changed the thylakoid and chlorophyll contents	[64]
79.0 ± 8.0	0.05–2 mg/L	*Lemna minor*	Caused decays on growth rate and fronds per colony; Induced oxidative stress	[65]
3.1–8.7	20, 200, 2000 mg/kg	Wheat (*Triticum aestivum* L.)	Caused lower biomass, shorter plant height, and lower grain weight; Decreases in the contents of micronutrients (Fe, Cu, and Zn); Decreased the contents of arginine and histidine	[61]
17.2 ± 0.3	0, 1, 10 and 30 mg/L (soybean); 0, 0.1, 0.5, 1 mg/L (rice)	Soybean; Rice	Significantly reduced plant biomass; Increased the malondialdehyde and H_2_O_2_ contents of leaves	[49]
12.9 ± 9.1	0.01, 0.05, 0.1, 0.5, 1 mg/L	*Capsicum annuum*	Decreased plant height and biomass; Causued a significant increase in total cytokinins in the leaves	[12]
20	1000, 3000 μM	*Pisum sativum*	Declined growth, photosynthetic pigments, and chlorophyll fluorescence; Inhibited activities of glutathione reductase (GR) and dehydroascorbate reductase (DHAR).	[66]
61.2 ± 33.9 (AgNP-Citrate); 9.4 ± 1.3 (AgNP-PVP); 5.6 ± 2.1 (AgNP-CTAB^c^)	25, 50, 75, 100 μM	*Allium cepa*	Caused oxidative stress; Led to strong reduction of the root growth	[67]
20	5, 10, 20 mg/L	*Allium cepa*	Induced various chromosomal aberrations in both mitotic and meiotic cells	[68]
200–800	1 mg/L	*Trigonella foenum-graecum* L.	Enhancement in plant growth and diosgenin synthesis	[69]
20	10–150 mg/L	*Arabidopsis thaliana*	Inhibited root gravitropism; Reduced auxin accumulation in root tips; Downregulated expression of auxin receptor-related genes	[70]
47	1, 3 mM	Mustard (*Brassica* sp.)	Declined growth of *Brassica* seedlings; Induced oxidative stress	[71]
35, 73	10 mg/L	Cucumber (*Cucumis sativus*); Wheat (*Triticum aestivum* L.)	Teduced growth; Upregulation of genes involved in the ethylene signalling pathway	[72]
5–50	800 µg/kg	*Vicia faba* L.	Declined germination; Decreased shoot and root length; Retarded the process of nodulation; Caused early senescence of root nodules	[73]
20	0, 2, 10, 20 mg/L	Potato (*Solanum tuberosum* L.)	Total reactive oxygen species (ROS) and superoxide anions were increased; Significant increases in the activities ofsuperoxide dismutase (SOD), catalase, ascorbate peroxidase, and glutathione reductase (GR); Higher ion leakage and cell death	[74]
35–40	50, 75 mg/L	*Triticum aestivum*; *Vigna sinensis*; *Brassica juncea*	50 ppm treatment promoted growth and increased root nodulation in cowpea; Improved shoot parameters at 75 ppm in *Brassica*	[75]
2	0, 125, 250, 500 mg/L	*Raphanus sativus*	Water content was reduced; Root and shoot lengths were reduced at 500 mg/L treatment; Significantly less Ca, Mg, B, Cu, Mn, and Zn	[76]
41	100–5000 mg/L	*Arabidopsis thaliana*	Reduced root length, leaf expansion and photosynthetic efficiency; Induced ROS accumulation; Induced Ca^2+^ in cytoplasm, inhibited plasma membrane K^+^ efflux and Ca^2+^ influx currents	[77]
100	50–100 μM	*Arabidopsis thaliana*	Accumulated more amino acids	[78]
10	1, 2, 5, 8, 10 mg/L	*Wolffia globosa*	Caused oxidative damage, higher malondialdehyde (MDA) content and an upregulation of SOD activity; Decreased contents of chlorophyll a, carotenoids and soluble protein	[79]
20	5 mg/L	*Arabidopsis thaliana*	111 genes were unique in AgNPs and enriched in three biological functions: response to fungal infection, anion transport, and cell wall/plasma membrane related.	[80]
10, 20, 40, 80	0.2 μg/L	*Arabidopsis thaliana*	Inhibition of root hair development; Repressed transcriptional responses to microbial pathogens, resulting in increased bacterial colonization	[81]
60–100 (Ag_2_S-NPs); 15–20 (AgNPs)	0–20mg/L (Ag_2_S-NPs); 0–1.6 mg/L (Ag-NPs)	Cowpea (*Vigna unguiculata* L. *Walp.*); Wheat (*Triticum aestivum* L.)	Ag_2_S-NPs reduced growth by up to 52%; Ag accumulated as Ag_2_S in the root and shoot tissues after exposed to Ag_2_S-NPs	[82]
20	75–300 μg/L	*Arabidopsis thaliana*	Prolonged vegetative and shortened reproductive growth; Decreased germination rates of offspring	[40]
6, 20	0.5, 5, 10 mg/L	*Spirodela polyrhiza*	Dose dependent increase in levels of ROS, SOD, peroxidase, and the antioxidant glutathione content; Chloroplasts accumulated starch grains and had reduced intergranal thylakoids.	[83]
20	0, 0.2, 0.5, 1 mg/L	*Arabidopsis thaliana*	Significantly reduced total chlorophyll and increased anthocyanin content; Increased lipid peroxidation; a dose-dependent increase in ROS production; Significant upregulated the expression of sulfur assimilation, glutathione biosynthesis, glutathione *S*-transferase, and glutathione reductase genes	[84]
20	0, 0.2, 0.5, 1 mg/L	*Oryza sativa* L.	Significant reduction in root elongation, shoot and root fresh weights, total chlorophyll, and carotenoids contents; Caused significant increase in H_2_O_2_ formation and lipid peroxidation in shoots and roots, increased foliar proline accumulation, and decreased sugar contents; Caused a dose dependent increase in ROS generation; Changes in mitochondrial membrane potential in the roots of seedlings	[53]
8, 45, 47	2–100 μM	*Arabidopsis thaliana*	Induced root growth promotion (RGP) and Cu/Zn superoxide dismutase (CSD2) accumulation; Inhibited ethylene (ET) perception and could interfere with ET biosynthesis	[85]
20,30–60, 70–120, 150	0.1, 1, 10, 100, 1000 mg/L	*Oryza sativa* L.	Seed germination and seedling growth were decreased	[86]
20, 40, 80	67–535 μg/L	*Arabidopsis thaliana*	Inhibited seedling root elongation; AgNPs were apoplastically transported in the cell wall and aggregated at plasmodesmata	[39]
20	5–25 mg/L	*Arabidopsis thaliana*	Upregulation of stress related genes, downregulation of pathogen and hormonal stimuli genes; Oxidative stress	[51]
10	0.2, 0.5, 3 mg/L	*Arabidopsis thaliana*	Root growth inhibition; Disrupted the thylakoid membrane structure and decreased chlorophyll content; Caused alteration of transcription of antioxidant and aquaporin related genes	[55]
11 ± 0.7 (Citrate)	0.05, 0.1, 1, 18.3, 36.7, 73.4 mg/L	*Zea mays Brassica oleracea*	Cell erosion in maize root apical meristem	[87]
18.34	0.30–60 mg/L	*Oryza sativa* L.	Damage the cell morphology and its structural features; Total soluble carbohydrates significantly declined; Caused production of the ROS and local root tissue death	[88]
10	0.5, 1.5, 2.5, 3.5, 5 mg/kg	*Triticum aestivum*	Teduced the length of shoots and roots; Caused oxidative stress in roots; Induced expression of a metallothionein gene involved in detoxification	[52]
10–15	0, 100, 1000 mg/L	Tomatoes (*Lycopersicon esculentum*)	Significant decreases in root growth; Decreased chlorophyll contents and Higher SOD activity; Less fruit productivity,	[89]
5, 10, 25	0.01–100 mg/L	*Arabidopsis thaliana*; poplars	Stimulatory effect on root elongation, fresh weight, and evapotranspiration at sublethal concentrations; Toxicity increased with decreasing AgNPs size	[90]
20 (AgNP-PVP) 6 (AgNP-GA^d^)	1, 10, 40 mg/L	Eleven species of wetland plants	40 mg/L AgNPs-GA exposure significantly reduced the germination rate of three species and enhanced the germination rate of one species.	[91]
<100	250, 750 mg/L	*Cucurbita pepo*	Reduction in plant biomass and transpiration	[92]
5–25	0, 5, 10, 20, 40 mg/L	*Phaseolus radiates*; *Sorghum bicolor*	Inhibition of plant growth	[58]
<100	0, 100, 500 mg/L	*Cucurbita pepo*	Decreased rate of transpiration	[93]
60	12.5, 25, 50, 100 mg/L	*Vicia faba*	Increased the number of chromosomal aberrations and micronuclei, and decreased the mitotic index	[94]
190–1100	0, 25, 50, 100, 200 or 400 mg/L	*Brassica juncea*	Increase in root length and increase in vigor index; Improved photosynthetic quantum efficiency and higher chlorophyll contents; Induced the activities of antioxidant enzymes, resulting in reduced reactive oxygen species levels	[95]
20, 100	5 μg/L	*Lemna minor* L.	Inhibition of plant growth	[57]
25	50, 500, 1000 mg/L	*Oryza sativa*	Broken the cell wall and damaged the vacuoles of root cells	[96]
24–55	0–80 mg/L	*Allium cepa*	Induced cell death and DNA damage through generation of ROS	[97]
<100	100 ppm	*Allium cepa*	Disturbed mitosis, reduction in mitotic index, declined metaphase, sticky chromosome, disintegration and breakdown of cell wall	[98]
20	100 mg/L	Green asparagus	Higher ascorbic acid and total chlorophyll contents	[99]

^a^ AgNP-B: AgNPs without surface coating; ^b^ AgNP-PVP: polyvinylpyrrolidone-coated silver nanoparticles; ^c^ AgNP-CTAB: cetyltrimethylammonium bromide-coated silver nanoparticles; ^d^ AgNP-GA: gum arabic-coated silver nanoparticles.

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
