# Peer review of "Impacts of Silver Nanoparticles on Plants: A Focus on the Phytotoxicity and Underlying Mechanism"

_ijms, 2019, doi:10.3390/ijms20051003_

Round 1

Reviewer 1 Report

A view of the “Impacts of silver nanoparticles on plants: a focus on the phytotoxicity and the underlying mechanism” by An Yan and Zhong Chen analyzed the current information on the toxic effects of AgNPs on higher plants. The review has a thoughtful structure, written in the light of the new data presented in the literature. It may be accepted for publication after technical revision.

Technical improvements.

1. In many cases, references in the text are indicated as Geisler-Lee et al., (2013) and at the end of a sentence [39]. Follow: Geisler-Lee et al. [39].

2. The reference list should have DOI.

3. Please, add an explanatory legend to Fig. 1.

4. Please, explain what AgNP-P and AgNP-S mean and what features in the action of biosynthesized and commercial AgNPs were found in the cited work.

At the end, the authors may ignore this comment that they do not consider the potential toxic effect of AgNPs as a result Ag(I) interfere to the plant copper metabolism because Ag(I) is electronically like Cu(I). In particular, the active center of plastocyanin contains Cys and His residues, which coordinate Ag (I) too. Metabolic replacement of copper by silver in plastocyanin will lead to a violation of photosynthesis. The opinion of the authors on such a mechanism of the toxic action of the AgNPs would contribute to the originality of the review.

Author Response

Response to comments: Reviewer 1: 1. In many cases, references in the text are indicated as Geisler-Lee et al., (2013) and at the end of a sentence [39]. Follow: Geisler-Lee et al. [39]. Response: We have corrected the way of citation as suggested. 2. The reference list should have DOI. Response: We have added DOI to the reference list accordingly. 3. Please, add an explanatory legend to Fig. 1. Response: We have added an explanatory legend to Figure 1 on Page 6. 4. Please, explain what AgNP-P and AgNP-S mean and what features in the action of biosynthesized and commercial AgNPs were found in the cited work. Response: We have added explanation of AgNP-P and AgNP-S, and added description of the features of AgNP-P and AgNP-S according to the cited work on Page 7. 5. At the end, the authors may ignore this comment that they do not consider the potential toxic effect of AgNPs as a result Ag(I) interfere to the plant copper metabolism because Ag(I) is electronically like Cu(I). In particular, the active center of plastocyanin contains Cys and His residues, which coordinate Ag (I) too. Metabolic replacement of copper by silver in plastocyanin will lead to a violation of photosynthesis. The opinion of the authors on such a mechanism of the toxic action of the AgNPs would contribute to the originality of the review. Response: Thanks for the nice suggestion and we have added description of the mechanism through which ionic silver affects photosynthesis on Page 8.

Reviewer 2 Report

The authors presented a summarization of the impacts of silver nanoparticles on plants, as well as the toxicity mechanisms and tolerance mechanisms based on the researched and studies in the past decades. The review paper is generally well organized, with conclusions supported the by the references. However, sections presenting the toxic mechanisms could be made more clear with tables or graphs. And as stated several times in the paper, a greater concern would be on the further damage caused to the ecosystem and to human health through plants, which could be added into the paper as a section.

Minor problems includes: the spelling and grammar could be improved; and the fonts of texts in Figure 1 should be enlarged as they are barely readable now.

Author Response

Response to comments: Reviewer 2 1. Sections presenting the toxic mechanisms could be made more clear with tables or graphs. Response: We have prepared Figure 2 to illustrate AgNPs properties which contribute to the phytotoxicity of AgNPs on Page 12. 2. As stated several times in the paper, a greater concern would be on the further damage caused to the ecosystem and to human health through plants, which could be added into the paper as a section. Response: We have added a section describing the potential risk in human health posed by AgNPs on Page 14 as suggested. 3. Minor problems includes: the spelling and grammar could be improved; and the fonts of texts in Figure 1 should be enlarged as they are barely readable now. Response: We have enlarged the fonts of texts in Figure 1 on Page 6. And we have checked throughout and corrected all spelling and grammar mistakes found.
